# Extant interspecific hybridization among trematodes within the *Schistosoma haematobium* species complex in Nigeria

Oluwaremilekun G. Ajakaye[1,2]*, Elisha E. Enabulele[3], Joshua B. Balogun[4], Oyetunde T. Oyeyemi[5], Michael E. Grigg[1]*

1 Molecular Parasitology Section, Laboratory of Parasitic Diseases, NIAID, National Institutes of Health, Bethesda Maryland, United States of America, 2 Department of Animal and Environmental Biology, Adekunle Ajasin University, Akungba Akoko, Nigeria, 3 Disease Intervention and Prevention Program, Biomedical Research Institute, San Antonio, Texas, United States of America, 4 Department of Biological Sciences Federal University, Dutse, Nigeria, 5 Department of Biosciences and Biotechnology, University of Medical Sciences, Ondo, Nigeria

* oluwaremilekun.ajakaye@aaua.edu.ng (OGA); griggm@niaid.nih.gov (MEG)

## Abstract

### Background

Natural interspecific hybridization between the human parasite (*Schistosoma haematobium* [Sh]) and bovine parasites (*Schistosoma bovis* [Sb], *Schistosoma curassoni* [Sc]) is increasingly reported in Africa. We developed a multi-locus PCR DNA-Seq strategy that amplifies two unlinked nuclear (trans*ITS*, *BF*) and two linked organellar genome markers (*CO1*, *ND5*) to genotype *S. haematobium* eggs collected from infected people in Ile Oluji/Oke Igbo, Ondo State (an agrarian community) and Kachi, Jigawa State (a pastoral community) in Southwestern and Northern Nigeria, respectively.

### Principal findings

Out of a total of 219 urine samples collected, 57 were positive for schistosomes. All patients from Jigawa state possessed an *Sh* mitochondrial genome and were infected with a genetic profile consistent with an *Sh x Sb* hybrid based on sequences obtained at *CO1*, *ND5*, trans*ITS* and *BF* nuclear markers. Whereas samples collected from Ondo state were more varied. Mitonuclear discordance was observed in all 17 patients, worms possessed an *Sb* mitochondrial genome but one of four different genetic profiles at the nuclear markers, either admixed (heterozygous between *Sh x Sc* or *Sh x Sb*) at both markers (n = 10), *Sh* at *BF* and admixed at trans*ITS* (*Sh x Sc*) (n = 5), admixed (*Sh x Sc*) at *BF* and homozygous *Sc* at trans*ITS* (n = 1) or homozygous *Sh* at *BF* and homozygous *Sc* at trans*ITS* (n = 1).

### Significance

Previous work suggested that zoonotic transmission of *S. bovis* in pastoral communities, where humans and animals share a common water source, is a driving factor facilitating interspecific hybridization. However, our data showed that all samples were hybrids, with greater diversity identified in Southwestern Nigeria, a non-pastoral site. Further, one patient

**Data Availability Statement:** All sequences have been deposited in GenBank, with accession numbers provided within the main body of the

manuscript accordingly: transITS Accession # OQ559623-OQ559638 & OQ564407-OQ564445; CO1 Accession #OQ568654-OQ568695; ND5 Accession #OQ571658 -OQ571713; BF Accession #OR574883-OR574888.

**Funding:** This work was supported by an International Foundation of Science (IFS) grant I3-B-6522-1 to OGA, and by the Division of Intramural Research project (AI001018) within the National Institute of Allergy and Infectious Diseases (NIAID) at the National Institutes of Health (NIH) to MEG. OGA is a recipient of an African Postdoctoral Training Initiative (APTI) fellowship jointly funded by the NIH, African Academy of Sciences (AAS) and the Bill and Melinda Gates Foundation (BMGF). The funders had no role in study design, data collection and analysis, decision to publish, or preparation of the manuscript.

**Competing interests:** The authors have declared that no competing interests exist.

possessed an *S. bovis* mitochondrial genome but was homozygous for *S. haematobium* at *BF* and homozygous for *S. curassoni* at trans*ITS* supporting at least two separate backcrosses in its origin, suggesting that interspecific hybridization may be an ongoing process.

## Author summary

Interspecific hybridization between trematode parasites pose serious health risks to humans. Many systems have shown possible hybridization between different schistosome species. As evidence of natural hybridization between human *S. haematobium* and animal *S. bovis* or *S. curassoni* has grown in recent years, epidemiological surveys across potential hybrid zones are required, particularly in endemic African regions. According to several reports, indiscriminate human-animal water contact is a major factor contributing to hybridization of human and animal schistosomes. We collected and genotyped 57 samples from pastoral and non-pastoral communities in Kachi, Jigawa state, and Ile Oluji/Oke Igbo, Ondo state, Nigeria to screen for hybrids. In both sites, we found *Schistosoma* hybrids with mitonuclear discordance that supported repeated backcrossing between *S. haematobium*, *S. bovis*, and *S. curassoni*. Contrary to previous reports, *Schistosoma* hybrids appear to be widespread and not solely dependent on human-animal water interactions.

## Introduction

Schistosomiasis is a highly prevalent water transmitted disease classified by the World Health Organization (WHO) as neglected. It is the second most prevalent tropical disease and is caused by six major species of flat worms, specifically *Schistosoma mansoni*, *S. haematobium*, *S. japonicum*, *S. mekongi*, *S. guineensis*, and *S. intercalatum*. The first two species cause approximately 20 million infections in Africa and are associated with severe chronic health consequences in affected populations [1,2]. Many other schistosome species, including *S. bovis*, *S. curassoni*, *S. mattheei* are known to infect livestock especially in sub–Saharan Africa [3]. Nigeria has the highest number of schistosomiasis cases in the world, with over 26% of Nigerians requiring chemotherapy. Despite annual mass chemotherapy programs, the urogenital form of schistosomiasis, caused by *S. haematobium*, remains a significant public health problem, with a current national prevalence of about 10% [4].

In 2009, field evidence of hybridization between *S. haematobium* and the bovine schistosome *S. bovis* was reported for the first time in Africa based on typing strategies using the mitochondrial marker *Cox1* (*CO1*) and the trans*ITS* (*ITS*) nuclear marker, which includes the *ITS1*, *5.8S*, and *ITS2* region located within the ribosomal RNA gene array [5]. Since the 2009 report, hybrid *S. haematobium* infections are now widely reported in countries in Africa, including Nigeria [6,7], and are thought to be producing emergent zoonoses between human urogenital *S. haematobium* with intestinal species of livestock schistosomes that include *S. bovis*, *S. curassoni*, and *S. bovis x S. curassoni* hybrids [5,6]. How such interspecific hybridization impacts urogenital schistosomiasis transmission, virulence, and treatment drug efficacy remains enigmatic.

More recently, several genome sequencing projects investigated *S. haematobium* [8–10] and *S. bovis* [11] at whole genome resolution to infer genome ancestry. Hybrid ancestry was extant, but the proportion of the nuclear genome derived from *S. bovis* was limited, at only

8.2% [8] or 23% [10] within the *S. haematobium* genomes investigated. The absence of full genome hybrids was interpreted to suggest that interspecies hybridization between *S. haematobium* x *S. bovis* is ancient, that it has occurred only rarely, and that *S. bovis* derived genome blocks have been purified by ongoing hybridization with *S. haematobium*, the more relevant schistosome species naturally infecting humans. It was also hypothesized that hybridization was endemic and occurring where the cultural practices of humans and livestock sharing the same water bodies is common [5,12–14].

In Nigeria, nomadic farming and pastoralism are a common cultural practice, particularly in the northern regions of the country. Pastoral animals are grazed around rivers and streams which also serve as the main sources of water for the communities. Therefore, these regions exist as potential high-risk areas for hybridization between human and livestock schistosome species. To test this hypothesis, that interspecific *S. haematobium* hybrids possessing a mixed ancestry with bovine schistosomes are common in water bodies shared between humans and cattle, we sampled two ecologically distinct *S. haematobium* endemic communities in Southwestern and Northern Nigeria. The sampled community in southern Nigeria is agrarian in nature, devoid of livestock *i.e.*, agricultural activities are predominant and cattle rearing and/ or markets are absent whereas the community in the northern part of the country is pastoral. We report here the genetic profiles of *S. haematobium* samples collected from patients in the two locations using genetic markers anchored in the mitochondria (*CO1*, *ND5*) and two unlinked nuclear genome markers (trans*ITS*, *BF*). The identification of homozygous alleles derived from animal schistosome species in sites devoid of livestock supports the notion that ongoing hybridization and backcrosses with animal schistosome species is occurring, independent of human-animal freshwater usage. Our work highlights the need to apply modern high throughput techniques to screen for hybrids to assess the degree to which ongoing hybridization is occurring between animal and human schistosomes, and to identify the relevant hosts and infection reservoirs that are promoting these interactions.

## Results and discussion

### Field site data

Interspecific hybridization between animal and human schistosomes is thought to occur preferentially in pastoral sites where animals and people share water bodies. To assess the degree to which this was occurring in Nigeria, two ecologically distinct *S. haematobium* endemic communities in southwestern and northern Nigeria were sampled (Fig 1). In Ile Oluji/Oke Igbo, a site in which cattle rearing and/or cattle markets are absent within the community and no human-cattle water contact activities occur, we hereafter refer to as being devoid of livestock, 17 of 107 filters contained parasite eggs. DNA was extracted from these *S. haematobium* positive urine samples. Forty of 112 filters contained parasite eggs in the pastoral community in Kachi, and DNA was extracted from these *S. haematobium* positive urine samples. In total, the prevalence of urogenital schistosomiasis infection was higher at the pastoral site in Kachi (35.7%) compared to the agrarian site in Ile Oluji/Oke Igbo (15.9%). The age of the participants ranged from 3–19 years and differed significantly between the two locations (p<0.00001). The prevalence of infection was higher in males (55.8%) than females (45.2%), but this did not reach significance (Table 1). In both study sites, a heavy intensity of infection was recorded (≥50 eggs per 10 ml of urine) (Table 1).

### Schistosoma haematobium genotyping

To investigate whether interspecific schistosome hybrids were causing human infection at the two study sites, a Sanger sequencing strategy was employed against pools of eggs collected

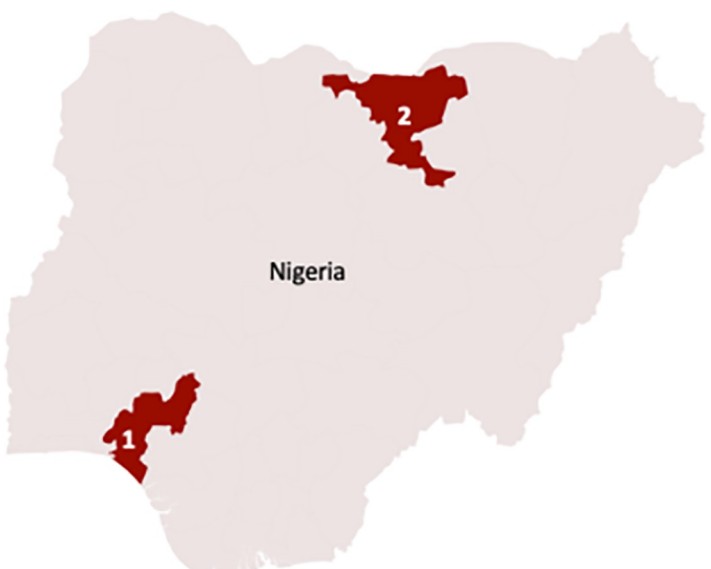

**Fig 1. Map of Nigeria showing the 36 states and the specific sampling locations for *S. haematobium* in Ondo State (1), Ile Oluji/Oke Igbo [latitude 5˚ 45N and 8˚ 15N and longitude 4˚ 30E], and Jigawa State (2), Kachi [latitude 11˚ 73' N and longitude 9˚ 33' E].** Map source: MAPSVG (CC BY 4.0 DEED; Attribution 4.0 International) at https://mapsvg.com/maps/nigeria.

from patient urine to determine the alleles present within the nuclear genome marker trans*ITS* and the mitochondrial genome marker *CO1*. In Kachi, all patients appeared to be infected by interspecific hybrids, the result of a cross between *S. bovis* x *S. haematobium* based on the two distinct sequences obtained at the trans*ITS* marker on chromosome 2 (Table 2). Of particular interest, a private SNP at nucleotide 502 (based on GenBank reference sequence MH014047) was identified that was unique to strains circulating in Kachi, this SNP has been identified previously at position 556 in the co-ordinate system used by these investigators [15] among *S. haematobium* or *S. bovis* samples collected from human infections (S2 Table). It was not possible to interpret the origin of the private SNP, *i.e.*, whether it was of *S. bovis* or *S. haematobium* ancestry. At *CO1*, all samples had an *S. haematobium* haplogroup I allele (Fig 2) [16]. One isolate possessed a single, private (or unique) SNP at *CO1* indicating that more than one haplotype was circulating among the patient samples (Fig 3).

**Table 1. *Schistosoma haematobium* prevalence and study population demographics in Ile Oluji/Oke Igbo (Ondo State) and Kachi (Jigawa State), Nigeria.**

| Variable | Category | Locations | | Total N (%) | P value |
|---|---|---|---|---|---|
| | | Ile Oluji/Oke Igbo N (%) | Kachi N (%) | | |
| Human Infection | Negative | 90 (84.1) | 72 (64.3) | 162 (74.0) | 0.0008* |
| | Positive | 17 (15.9) | 40 (35.7) | 57 (26.0) | |
| Infection Intensity | Heavy | 11 (64.7) | 17 (42.5) | 28 (49.1) | 0.1249 |
| | Light | 6 (35.5) | 23 (57.5) | 29 (50.9) | |
| Gender | Female | 54 (50.5) | 45 (40.2) | 99 (45.2) | 0.1262 |
| | Male | 53 (49.5) | 67 (59.8) | 120 (55.8) | |
| Age | < 10 | 73 (68.2) | 22 (19.6) | 95 (43.4) | < 0.00001* |
| | > 10 | 34 (31.8) | 90 (80.4) | 124 (56.6) | |

* P< 0.05, Percentages are in parenthesis

**Table 2.** *Schistosoma* **species by genetic marker for Ile Oluji/Oke Igbo and Kachi.**

| Isolate | Location | State | Source | Nuclear Markers | | Mitochondrial Markers | |
|---------|----------|-------|--------|-----------|------|------|------|
| | | | | transITS | BF | CO1 | ND5 |
| OD1 | Ile Oluji/Oke | Ondo | Human | S.h x S.c | S.h | S.b | S.b |
| OD2 | Ile Oluji/Oke | Ondo | Human | S.h x S.c | S.h x S.b | S.b | S.b* |
| OD3 | Ile Oluji/Oke | Ondo | Human | S.h x S.c | S.h | n.d. | S.b |
| OD4 | Ile Oluji/Oke | Ondo | Human | S.h x S.c | S.h | S.b | S.b |
| OD5 | Ile Oluji/Oke | Ondo | Human | S.h x S.c | S.h | S.b | S.b |
| OD6 | Ile Oluji/Oke | Ondo | Human | S.h x S.c | S.h x S.b | n.d. | S.b |
| OD7 | Ile Oluji/Oke | Ondo | Human | S.h x S.c | S.h | S.b | S.b |
| OD8 | Ile Oluji/Oke | Ondo | Human | S.h x S.c | S.h x S.b | S.b | S.b |
| OD9 | Ile Oluji/Oke | Ondo | Human | S.h x S.c | S.h x S.b | S.b | S.b |
| OD10 | Ile Oluji/Oke | Ondo | Human | S.h x S.c | S.h x S.b | S.b | S.b |
| OD11 | Ile Oluji/Oke | Ondo | Human | S.h x S.c | S.h x S.b | S.b | S.b |
| OD12 | Ile Oluji/Oke | Ondo | Human | S.h x S.c | S.h x S.b | S.b | S.b |
| OD13 | Ile Oluji/Oke | Ondo | Human | S.h x S.c | S.h x S.b | S.b | S.b |
| OD14 | Ile Oluji/Oke | Ondo | Human | S.c | S.h x S.c | S.b | S.b |
| OD15 | Ile Oluji/Oke | Ondo | Human | S.h x S.c | S.h x S.b | n.d. | S.b |
| OD16 | Ile Oluji/Oke | Ondo | Human | S.h x S.c | S.h x S.b | S.b | S.b |
| OD17 | Ile Oluji/Oke | Ondo | Human | S.c | S.h | S.b | S.b |
| J1 | Kachi | Jigawa | Human | S.h x S.b* | S.h | S.h | S.h |
| J2 | Kachi | Jigawa | Human | S.h x S.b* | S.h | S.h | S.h |
| J3 | Kachi | Jigawa | Human | S.h x S.b* | S.h | S.h | S.h |
| J4 | Kachi | Jigawa | Human | S.h x S.b* | S.h | S.h | S.h |
| J5 | Kachi | Jigawa | Human | S.h x S.b* | S.h | S.h | S.h |
| J6 | Kachi | Jigawa | Human | S.h x S.b* | S.h | S.h | S.h |
| J7 | Kachi | Jigawa | Human | S.h x S.b* | S.h | S.h | S.h |
| J8 | Kachi | Jigawa | Human | S.h x S.b* | S.h | S.h | S.h |
| J9 | Kachi | Jigawa | Human | S.h x S.b* | S.h | S.h | S.h |
| J10 | Kachi | Jigawa | Human | S.h x S.b* | S.h | S.h | S.h |
| J11 | Kachi | Jigawa | Human | S.h x S.b* | S.h | S.h | S.h |
| J12 | Kachi | Jigawa | Human | S.h x S.b* | S.h | S.h | S.h |
| J13 | Kachi | Jigawa | Human | S.h x S.b* | S.h | S.h | S.h |
| J14 | Kachi | Jigawa | Human | S.h x S.b* | S.h | S.h | S.h |
| J15 | Kachi | Jigawa | Human | S.h x S.b* | S.h x S.b | S.h | S.h |
| J16 | Kachi | Jigawa | Human | S.h x S.b* | S.h | S.h | S.h |
| J17 | Kachi | Jigawa | Human | S.h x S.b* | S.h | S.h | S.h |
| J18 | Kachi | Jigawa | Human | S.h x S.b* | S.h | S.h | S.h |
| J19 | Kachi | Jigawa | Human | S.h x S.b* | S.h | S.h | S.h |
| J20 | Kachi | Jigawa | Human | S.h x S.b* | S.h | S.h | S.h |
| J21 | Kachi | Jigawa | Human | S.h x S.b* | S.h | S.h | S.h |
| J22 | Kachi | Jigawa | Human | S.h x S.b* | S.h | S.h | S.h |
| J23 | Kachi | Jigawa | Human | S.h x S.b* | S.h | S.h | S.h |
| J24 | Kachi | Jigawa | Human | S.h x S.b* | S.h | S.h | S.h |
| J25 | Kachi | Jigawa | Human | S.h x S.b* | S.h | S.h | S.h |
| J26 | Kachi | Jigawa | Human | S.h x S.b* | S.h | S.h | S.h |
| J27 | Kachi | Jigawa | Human | S.h x S.b* | S.h | S.h | S.h |
| J28 | Kachi | Jigawa | Human | S.h x S.b* | S.h | S.h | S.h |

*(Continued)*

**Table 2.** (Continued)

| Isolate | Location | State | Source | Nuclear Markers | | Mitochondrial Markers | |
|---------|----------|-------|--------|-----------------|---|-----------------------|---|
| | | | | transITS | BF | CO1 | ND5 |
| J29 | Kachi | Jigawa | Human | S.h x S.b* | S.h | S.h | S.h |
| J30 | Kachi | Jigawa | Human | S.h x S.b* | S.h | S.h | S.h |
| J31 | Kachi | Jigawa | Human | S.h x S.b* | S.h | S.h | S.h |
| J32 | Kachi | Jigawa | Human | S.h x S.b* | S.h | S.h | S.h* |
| J33 | Kachi | Jigawa | Human | S.h x S.b* | S.h | S.h | S.h |
| J34 | Kachi | Jigawa | Human | S.h x S.b* | S.h | S.h | S.h |
| J35 | Kachi | Jigawa | Human | S.h x S.b* | S.h | S.h | S.h |
| J36 | Kachi | Jigawa | Human | S.h x S.b* | S.h | S.h | S.h |
| J37 | Kachi | Jigawa | Human | S.h x S.b* | S.h | S.h | S.h |
| J38 | Kachi | Jigawa | Human | S.h x S.b* | S.h | S.h | S.h |
| J39 | Kachi | Jigawa | Human | S.h x S.b* | S.h | S.h | S.h |
| J40 | Kachi | Jigawa | Human | S.h x S.b* | S.h | S.h | S.h |

The following 2 letter codes were used to identify the species type sequenced for each genetic marker:

S.h = S. haematobium; S.b = S. bovis; S.c = S. curassoni; S.h x S.b = S. haematobium x S. bovis; S.h x S.c = S. haematobium x S. curassoni;

* presence of a private SNP;

n.d. = failed PCR/sequencing reaction.

transITS Accession # OQ559623-OQ559638 & OQ564407-OQ564445; CO1 Accession #OQ568654-OQ568695; ND5 Accession #OQ571658 -OQ571713; BF Accession #OR574883-OR574888

In Ile Oluji/Oke Igbo, an endemic site devoid of livestock, all patients likewise appeared to be infected with interspecific hybrids. At the *CO1* marker, all samples possessed alleles that branched with *S. bovis* (Table 2). Phylogenetic analysis at *CO1* showed that infected patients possessed one of 4 unambiguous sequences that branched separately, but clustered tightly with other *S. bovis* sequences, including those from a cow in Nigeria (this study) and from the public domain, that resolved as a monophyletic group and was distinguished from other human and livestock schistosomes within the *S. haematobium* species complex with good bootstrap support, including *S. curassoni*, *S. mattheei*, *S. intercalatum*, and *S. guineensis* (Fig 2). At *CO1*, one allele was dominant, and the other 3 alleles possessed a single, private SNP indicating that at least 4 haplotypes were circulating among the patient samples (Fig 3). Further, mitonuclear discordance was observed in all 17 samples, worms possessed an *S. bovis* mitochondrial ancestry, but one of 2 different genetic profiles based on the alleles present at the nuclear trans*ITS* marker, either admixed (heterozygous between *S. haematobium x S. curassoni*) or homozygous for *S. curassoni*. The presence of alleles from zoonotic livestock-specific schistosomes in two patients (OD14 & OD17), the result of hybridization between *S. bovis* x *S. curassoni* with no evidence of *S. haematobium* was noteworthy but has been reported previously in Niger [17]. The other 15 samples, however, were consistent with livestock:human schistosome hybrids. Our data show that an array of hybrid livestock:human schistosomes are circulating in Nigeria, and they are causing infections in agrarian (non-pastoral) sites, which are devoid of livestock. This was not expected, because the PCR-DNA-seq analysis was carried out on pools of eggs from different patient samples. The prevailing consensus presupposes greater parasite diversity in endemic areas, often referred to as the genetic mixing bowl hypothesis [6] in which continuous re-infection in an infected host pool is thought to result in an increased number of paired

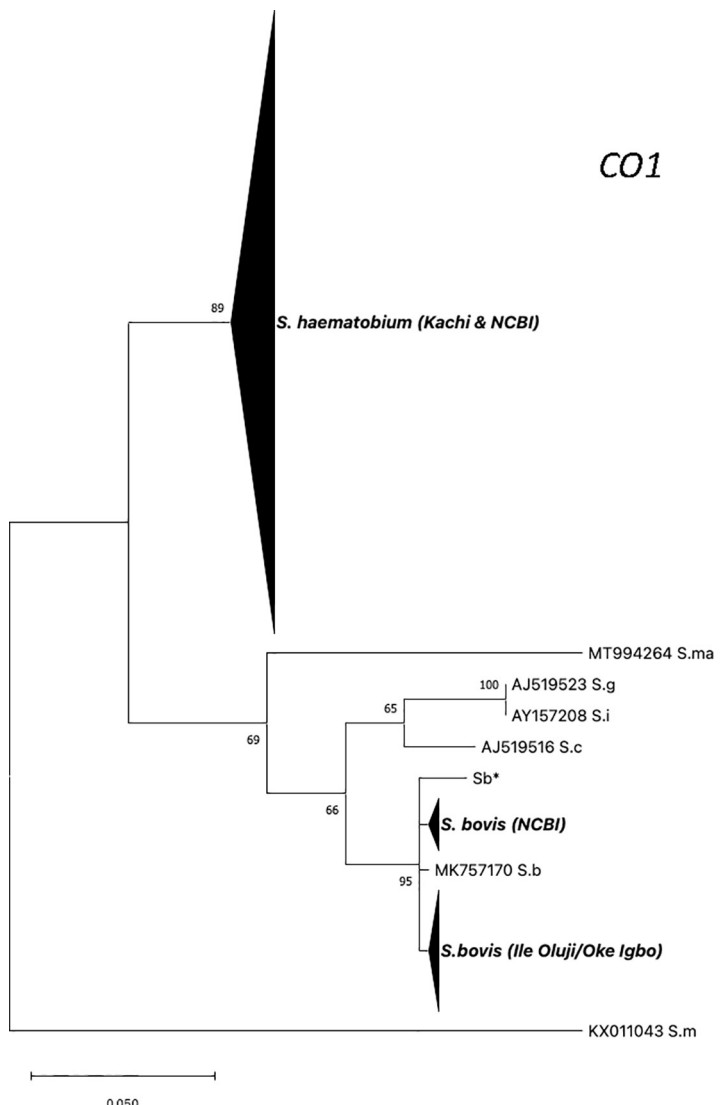

**Fig 2. Maximum likelihood tree based on CO1 sequences obtained from _S. haematobium_ samples collected in Oluji/Oke Igbo and Kachi in Nigeria.** Reference sequences from worms within the _S. haematobium_ species complex were added for comparative purposes, and are comprised of published sequences retrieved from GenBank. The OC1 sequence from _S. mansoni_ (KX011043) was used as the outgroup. (Sb*—this study, _S. bovis_ worm from cow recovered from an abattoir in Nigeria); _S.b = S. bovis; S.m = S. mansoni; S.c = S. curassoni; S.ma = S. mattheei; S.g = S. guineensis; S.i = S. intercalatum._

schistosomes within a patient, and greater parasite diversity, as previously reported in Niger [17]. Hence, the expectation was that sequences derived from a "pool" of eggs would be quite variable, with different peak heights, and changes in heterozygosity profiles between independent replicates. However, this was not the case (S2 Table). The electropherograms derived from PCR-DNA sequencing pools of eggs at the _transITS_ either produced a single homozygous sequence type across independent PCR reactions (samples OD14 and OD17 possessed only _S. curassoni_ alleles), or sequences that possessed equal peak heights for all bi-alleleic SNPs at sites of heterozygosity, that repeated with high confidence between independent PCR replicates, which supports a strictly "hybrid" interpretation (S2 Table). The origin of these schistosome hybrids is unclear as the two markers applied are insufficiently resolved to differentiate

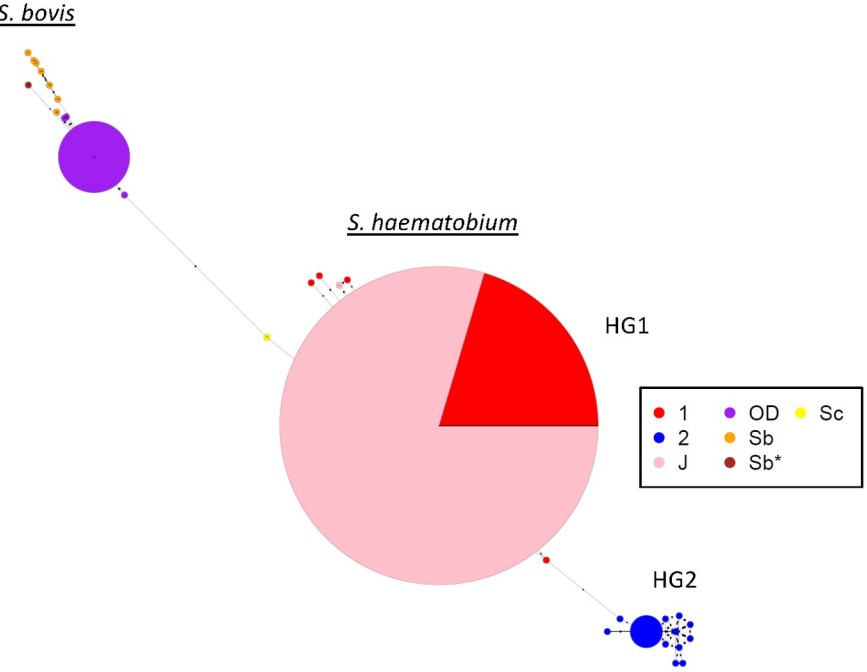

**Fig 3. Haplotype network at *CO1*.** *S. haematobium CO1* alleles circulating among samples recovered from patients in Kachi, Jigawa state (J) belonged to the *S. haematobium* group 1 (HG1) haplotype (pink). Red–*S. haematobium* haplogroup 1 reference sequences obtained from GenBank. For reference, *S. haematobium* group 2 (HG2) haplotypes (blue) are also depicted for sequences obtained from GenBank. *S. bovis CO1* alleles circulating among samples recovered from patients in Oluji/Oke Igbo, Ondo state (OD) were distinct (purple) from sequences obtained from GenBank annotated as *S. bovis* (orange). Sc represents the *CO1* allele obtained for *S. curassoni* (yellow). Sb* represents the allele identified at *CO1* for the *S. bovis* isolate recovered from an abattoir in Nigeria, that served as a control sequence for *S. bovis*.

between an ancestral introgression versus a contemporary hybridization event in their origin. However, recent work at genome resolution, among naturally occurring animal schistosome hybrids supports ongoing hybridization in the origin of new zoonotic lineages which have important veterinary and public health implications [18].

## Bravo-Figey (BF) domain-containing nuclear and Nad5 (ND5) mitochondrial markers

To better understand the true genetic diversity present, and to further resolve the origin of the *S. bovis* x *S. curassoni* hybrids, we developed two additional genetic markers. The first, *BF*, a cell adhesion molecule (CAM) that belongs to the N-CAM transmembrane proteins, represents an unlinked marker located on chromosome 7. The gene has seven exons, is 4164 nucleotides in length, and PCR primers were designed in a nested configuration to amplify a polymorphic fragment within exon 2 that is 408 nucleotides in length. The genus-specific primers amplify all *Schistosoma* species with equal efficiency. DNA sequencing was next used to distinguish between the species, of which 29, 14, 9 and 3 SNPs separate the *S. haematobium* allele from that of *S. mansoni*, *S. bovis*, *S. curassoni* and *S. mattheei* reference sequences (GenBank Accession IDs OR574883-OR574888), respectively (Fig 4). In Kachi, all samples but one possessed an unambiguous, homozygous *S. haematobium BF* allele whereas isolate J15 was heterozygous and had an *S. bovis* x *S. haematobium* allele (Table 2). In total, only two hybrid *Sb x Sh* genetic profiles were resolved in Kachi among the 4 markers applied (Table 2). In contrast,

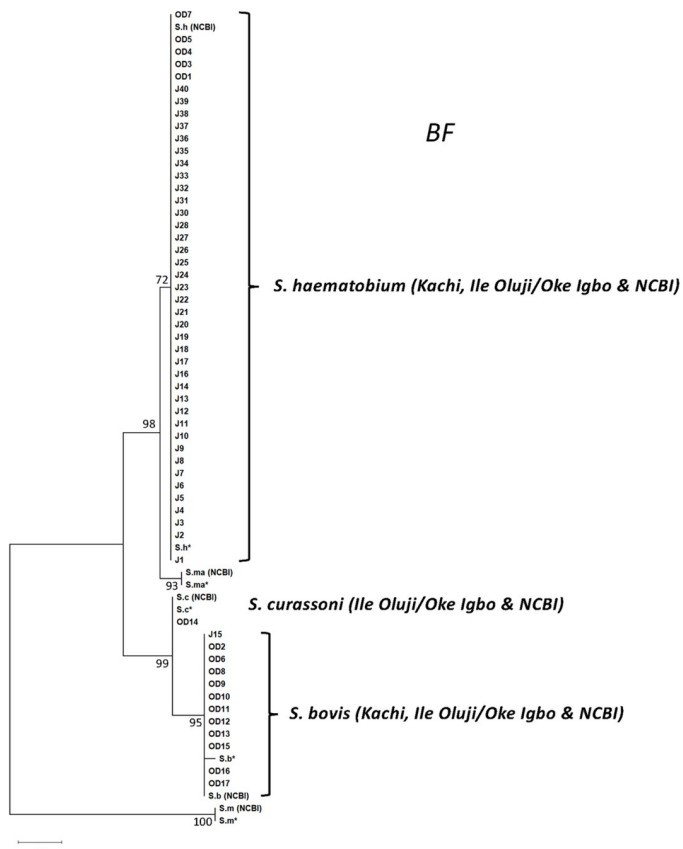

**Fig 4. Maximum likelihood tree at BF.** Phylogenetic analysis of *BF* alleles recovered from *S. haematobium* samples in Oluji/Oke Igbo, Ondo state and Kachi, Jigawa state in Nigeria were compared against reference sequences (S.m* = *S. mansoni*, S.ma* = *S. mattheei*, S.c* = *S. curassoni* and *S. bovis*; OR574883-OR574888). For samples that were heterozygous, only non- *S.haematobium* alleles are shown in the tree. Hence, for sample OD14, only the *S. curassoni* allele is depicted for the Sh x Sc alleles present at *BF*. Likewise, for OD2,6,8–13, 15–16, and J15, only the *S. bovis* allele is depicted for the Sh x Sb alleles present at *BF*. All other samples were homozygous for *S. haematobium* alleles at *BF*.

in Ile Oluji/Oke Igbo, 3 allelic types were resolved at *BF*, either a homozygous *S. haematobium* allele in 6 of 17 samples, a heterozygous admix of *S. haematobium* x *S. bovis* alleles in 10 of 16 samples, and one heterozygous admix of *S. haematobium* x *S. curassoni* in the final sample (Table 2). Importantly, in the two samples (OD14, OD17) that represented *Sb* x *Sc* livestock: livestock hybrids when assessed at just *CO1* and trans*ITS* genotyping markers, inclusion of the *BF* marker established that the genotypes for these two samples were in fact hybrid *Sb x Sc* live-stock:human *Sh* recombinants that appear to have undergone repeated interspecific hybridizations, including at least two separate backcrosses, in the origin of these human infective schistosomes. The presence of such hybrid *Sb* x *Sc* livestock:human *Sh* genotypes have been described previously and support a model whereby no real or perceived barrier of genetic hybridization exists between livestock and human species of trematodes within the *S. haematobium* species complex [17]. For the majority of the Ile Oluji/Oke Igbo samples, a cross between an *S. haematobium* male with a hybrid *S. bovis x S. curassoni* female would be sufficient to produce the genotype resolved, specifically a maternally inherited *S. bovis* mitochondrial genome that is heterozygous with an *Sh* and either an *Sb* or *Sc* allele at the nuclear markers. Further, in this agrarian community, in the absence of livestock, it may also suggest that people are playing an important role in the origin and evolution of these human infective hybrids. It will be

important to prospectively sample at these collection sites and genotype, using an increased number of genetic markers or at whole genome resolution, individual miracidia isolated from patient egg samples to determine the extent to which mixed infections are occurring that could promote cross-pairing among the various species within the *S. haematobium* species complex.

Finally, recombination and rapid *in situ* evolution has been shown to occur in mitochondrial genomes for a variety of parasitic pathogens during interspecific mating [19] or during passage [20]. We developed pan-genus PCR primers in a nested configuration to amplify the *ND5* gene within the mitochondrial genome. The genus-specific primers amplified all *Schistosoma* species with equal efficiency. When applied against all samples, no interspecific recombination was resolved, and the *ND5* sequence types were in linkage with the sequence type resolved at *CO1* confirming that the samples from Kachi had inherited an *S. haematobium* mitochondrial genome, and those from Ile Oluji/Oke Igbo had inherited an *S. bovis* mitochondrial genome (Table 2). Two samples, one each from Kachi and Ile Oluji/Oke Igbo, each possessed a single private SNP at *ND5*, increasing the number of haplotypes resolved among the patient samples (Table 2 and S2 Fig) from Nigeria.

## Conclusion

Although the *CO1* and trans*ITS* gene markers are reasonably proficient at detecting hybrids, the inclusion of additional markers allowed for a better resolution of complex interspecies hybridization events, including livestock only hybrids, as evidenced herein that was consistent with a cross between a human *Sh* strain with an *Sc* x *Sb* livestock:livestock hybrid that infected two individuals. Hence there is a need to develop more markers and apply them using modern high throughput methods to more precisely determine the extent and number of circulating interspecific hybrids to prioritize these samples for whole genome sequencing. We propose a multilocus typing scheme involving linked and unlinked nuclear gene markers located on the seven *S. haematobium* chromosomes to determine the extent of hybridization that is occurring in schistosomiasis-endemic settings. We also provide compelling evidence that admixed *S. haematobium*:livestock schistosomes are widespread and their origin and transmission may be less dependent on human/cattle contact than previously envisaged. Investigating the genetic diversity and degree of interspecific *S. haematobium* hybrids circulating in snail vector and other animal reservoir hosts (for example, rodents) may provide useful insight in the transmission and origin of these interspecific hybrids. It will also be important to genotype individual miracidia derived from infected individuals to assess the degree of mixed infections, and to assess whether cross-pairings with animal:human schistosomes is common, and occurring *in situ* in infected people.

Finally, there is a need for whole genome sequencing analysis of multiple *S. haematobium* samples from areas where livestock schistosomes are absent. It will be important to screen isolates from Zanzibar or Madagascar where no *S. bovis* ancestry has been recorded. This will provide the genomic backbone to compare whole genome sequences of *S. haematobium* populations from different geographical regions, in order to map introgressed and genomic regions that are promoting disease transmission and host range specificity.

## Materials and methods

### Ethics statement

This study was conducted under the National ethical permit and protocol numbers NHREC/01/01/2007–30/10/2020 and NHREC/01/01/2007–29/03/2022B obtained from the National Health Research Ethics Committee (NHREC), Department of Health Planning, Research and Statistics, Federal Ministry of Health, Nigeria. The study was conducted in rural communities

where the villagers provided only oral, not written, consent. After verbal consent was provided by parents and guardians, about 20ml of urine was collected in specimen bottles from a total number of 219 participants ranging in age from four to fourteen years. Participants found to be infected were contacted and treated with a single dose of praziquantel according to WHO's recommendation. Also, the investigation report was made available to the public health officer in each of the local government areas studied.

## Study sites

Ile Oluji/Oke Igbo is an administrative area of Ondo state situated in the tropical rainforest belt of Southwestern Nigeria. The area which lies between latitude 5˚ 45N and 8˚ 15N and longitude 4˚ 30E is made up of fourteen villages. The climate is humid with small seasonal and daily variations in rainfall. The rainfall is concentrated during the months of May to October with a short break in August and considerable variations from year to year. The area is surrounded by many rivers, such as Owena, Aigo, Esinmu, Iyire, Ogburu, Oni and Awo. The inhabitants of the study area are primarily farmers, engaging in small to medium scale production of both cash and arable crops like yam, cassava, kolanut, cocoa and maize among others [21] (Fig 1).

Kachi is a village situated in Northern Nigeria under Dutse Local Government Area of Jigawa State (latitude 11˚ 73' N and longitude 9˚ 33' E). Dutse is a city located in Northern Nigeria and is the capital of Jigawa State. The population is predominantly the Hausa and Fulani tribes practicing agriculture and pastoralism as the main source of livelihood. The climate of the area is tropical wet and dry, and the vegetation type is Sudan Savannah despite its rocky topography typical of Dutse. Over the course of the year, the temperature typically varies reaching highs of 39˚C [22] (Fig 1).

## Parasitological analysis

In the laboratory, urine samples were processed using the sedimentation and filtration technique. Briefly, urine samples were allowed to settle for one hour after which 10ml of urine was drawn from the bottom of the specimen bottle and filtered using Sterlitech 13 mm polycarbonate screen membrane filters (https://www.sterlitech.com/schistosome-test-kit.html) for collection of eggs. Filters were examined under the microscope for the presence of eggs and filters containing *S. haematobium* eggs were preserved by placing in absolute ethanol for subsequent molecular analysis. Identification of *S. haematobium* eggs was based on their characteristic terminal spine, and both the prevalence and infection intensity were determined by microscopy. Infection intensity was determined as the number of eggs detected per 10 ml of urine (eggs/10 ml). Light infection was categorized as 1–49 eggs/10 ml and heavy infections $\geq$50 eggs/10 ml [23]. Statistical analysis was done using the Fisher exact test to compare the prevalence and intensity of infection with demographic variables.

## Molecular analysis

Filters were removed from ethanol and air dried for thirty minutes. DNA extraction was carried out using QAIGEN Blood and Tissue kit following the manufacturers protocol (www.qiagen.com). Briefly, the filters were incubated at 56˚C in lysis buffer overnight. The filters were removed, 200 μl Buffer AL and ethanol were added to the lysate and mixed thoroughly by vortexing. The lysate was passed through the Qiagen columns and washed twice. Isolated DNA was eluted in 20ul of molecular grade water and stored at -20˚C.

Additionally, DNA from five *Schistosoma* species were used as positive controls *(S. mansoni, S. curassoni, S. bovis, S. mattheei,* and *S. haematobium)* that were PCR-amplified twice

and Sanger sequenced at all 4 markers used in this study, the sequence data was identical both times. Specifically, genomic DNA was extracted from one sample each of *S. mattheei*, and *S. curassoni* worms obtained from the Schistosomiasis Collection at the Natural History Museum (SCAN), UK [24] and an *S. bovis* worm (extracted from a cow in Nigeria) as described for filters above. Genomic DNA was obtained for a single sample each of *S. mansoni* and *S. haematobium* from the Schistosome Resource Centre, Biomedical Research Institute, USA. A partial region of the mitochondrial cytochrome oxidase subunit 1 (*CO1*) and the complete nuclear ribosomal internal transcribed spacers (trans*ITS*) gene regions was amplified for each sample using published primers Cox1_schist_5′ (5′-TCTTTRGATCATAAGCG-3′), Cox1_schist_3′ (5′-TAATGCATMGGAAAAAAACA-3′) and ETTS1 (5′-TGCTTAAGTTCAGCGGGT-3'), ETTS2 (5′-TAACAAGGTTTCCGTAGGTGAA-3'), respectively [25, 26]. The cycling condition for the PCR was as described by Lockyer et al, 2003 [25]. PCR amplicons were visualized on agarose gel prior to beads purification and sanger sequencing. To screen for the existence of mixed infections and to further resolve the species present, we developed two additional markers, the mitochondrial *Nad5* (*ND5*) and nuclear *Bravo Figey* domain containing protein (*BF*) gene markers. Our aim was to identify an additional linked mitochondrial marker (there is evidence in kinetoplastids for recombination within the mitochondrial genome, so we developed a linked marker here to assess that possibility). We used *ND5* as it has been shown to be polymorphic among different *Schistosoma* species (*S. mansoni*, *S. curassoni*, *S. bovis*, *S. mattheei*, *S. intercalatum*, *S. guineensis* and *S. haematobium*). We also developed a new, unlinked nuclear genome marker (*BF*) to assess incongruence, *BF* was targeted because it is a surface antigen that is polymorphic between different schistosome species within the *S. haematobium* species complex, and is under positive selection. In our previous work examining population genetic structures among various protists, surface antigen genes were identified as excellent genetic markers for phylogenetic analyses, largely because they are typically under diversifying pressure, and are sufficiently polymorphic. In this case, the *Bravo Figey* or *NrCAM* is a highly conserved protein which is part of the intracellular region of the neural adhesion molecule L1 proteins. The role of cell adhesion molecules in host parasite interactions has been recognized. They have been suggested as mediators of attachment of pathogens to host cells in protozoan parasites infections [27]. In *Trichobilharzia*, CAMs regulate cell adhesion to maintain their structure. They are upregulated in the schistosomula of *T. regenti* suggesting rapid growth and development of different organ structures within the definitive host [28]. Importantly the single gene copy *BF* marker (chromosome 7) is unlinked from trans*ITS* (chromosome 2), which allowed us to look for samples that were incongruent (*i.e.*, chromosomes of distinct ancestry), and thus support a meiotic process in the origin of the sample genotyped.

To develop the primers, we downloaded sequences from NCBI for each marker across all species of schistosome studied, performed an alignment in Geneious and identified regions of sequence conservation across all species that bracketed polymorphic stretches. PRIMER3 software was used to design primers that were pan-genus that bracketed polymorphic stretches that were species specific. A nested configuration of primers were developed at the nuclear gene marker, *Bravo Figey* domain containing protein gene (*BF*) as follows: EXT: FWD_5′-TGTATCACGCTGGCCATACT-3′ and REV_5′-CCACCTGCCATCAAACTCAC-3'; INT: FWD_5′-ACTAGATGGCAGATACGGACC-3′ and REV_5′-TAGTCCCCTTGAGGTTG TCG-3'. The following primer was developed at the mitochondrial *ND5* gene (5′-GGGTAAA AGTTGGAATTTGAGGG-3′ and 5′-CGCTTTAACCATCTGACCACC-3'). Both primers amplify a polymorphic region between *S. haematobium* and *S. bovis*. We validated the new primer sets by PCR amplification and DNA sequencing of the *ND5* and *BF* gene regions using five reference schistosome species (specifically *S. mansoni*, *S. curassoni*, *S. bovis*, *S. mattheei*, *S. intercalatum*, *S. guineensis* and *S. haematobium*) and obtained homozygous sequences for the

phylogenetic analysis. We used the following PCR cycling conditions for the *ND5* and *BF* primers, 3 minutes at 98˚C followed by 35 cycles each of 20 seconds at 98˚C, 15 seconds at 58˚C followed by 30 seconds at 72˚C, and a final elongation step at 72˚C for 1 minute. All PCRs were performed in a 25ul reaction using KAPA HIFI Taq reagents, 0.25ul of 50um forward and reverse primers and 1ul of DNA. 4ul of amplicons was visualized on agarose gel by gel electrophoresis. Amplicons were purified using Ampure XP beads and Sanger sequenced in both directions. Sequencing was performed on at least two independent PCR reactions for each marker across all samples. All Sanger sequencing analysis was done at the Rocky Mountain laboratory at the National Institute of Health, Maryland, USA.

## Phylogenetic analysis

All sequences were imported into Geneious vs 2022.2.1 for de novo assembly and trimming. Polymorphic positions in consensus sequences were crosschecked by visualizing the original chromatograms of the forward and reverse sequences. The NCBI nucleotide blast tool (https:// blast.ncbi.nlm.nih.gov) was used to initially confirm the identity of each consensus sequence. Several published *Schistosoma* species sequences were retrieved from the NCBI nucleotide database (S1 Table) to compare against the homozygous sequences we obtained for each of the typed specimens including *S. mansoni*, *S. curassoni*, *S. bovis*, *S. mattheei*, and *S. haematobium*. For sequences obtained from worms collected from patients that possessed sites of heterozygosity at the two nuclear markers, it was possible to phase the haplotypes based on the reference sequences, as this was the most parsimonious explanation for the origin of the heterozygosity and these alleles were used for the phylogenetic and network analyses performed in this study (Table 2). All sequences were aligned using MUSCLE program within the Geneious software prior to subsequent analysis. The molecular phylogenetic analysis of the sequence data for the four gene markers (*CO1* 392bp, *ND5* 336bp, trans*ITS* 880bp and *BF* 408bp) was implemented using MEGA X based on the best model (lowest Bayesian information criterion score) for each gene marker [29]. 1000 bootstrap replicates were collected using the Maximum Likelihood method to ascertain the reliability of the tree branches. Haplotype network was implemented for *CO1* sequence data in an R environment using the Pegas PE package [30].

## Supporting information

**S1 Fig. Maximum likelihood tree at the trans*ITS* marker for *S. haematobium* samples in Oluji/Oke Igbo (OD).** For samples OD14 and OD17, only an *S. curassoni* allele was recovered at trans*ITS*. All other samples were heterozygous and possessed an *S. haematobium* allele x *S. curassoni* allele. Only the *S. curassoni* allele from OD is depicted for comparison against reference alleles for the following: *S. bovis* Nig = *S. bovis* worm recovered from a cow in Nigeria. *S. bovis* (NCBI) = MW027649.1, MT158872.1, MW027648.1, MF776588.1, MF776589.1. *S. haematobium* (NCBI) = MH014047, GU257398, JQ397400, JQ397401, JQ397402, JQ397403, JQ397404, JQ397405, JQ397406, JQ397407, JQ397408, JQ397409, JQ397410, JQ397411, JQ397412, JQ397413, JQ397414. S.c = *S. curassoni* isolate from SEN (Senegal), GenBank Accession number MT580961. S. ma = *S. mattheei* isolate from ZIM (Zimbabwe), GenBank Accession number MW04687. S. m = *S. mansoni* isolate from GAB (Gabon), GenBank Accession number KX011041. (TIF)

**S2 Fig. Maximum likelihood tree at ND5.** Phylogenetic analysis of *ND5* alleles recovered from *S. haematobium* samples in Oluji/Oke Igbo (OD) and Kachi, (J) in Nigeria. Only *S. bovis* alleles were identified among samples recovered from Oluji/Oke Igbo (OD), whereas only *S.*

*haematobium* alleles were identified among samples recovered from Kachi, (J). * This study, (*S. bovis* worm from cow in Nigeria), S.g = *S. guineensis*, S.c = *S. curassoni*, S.m = *S. mansoni*. (TIF)

**S1 Table. *CO1* haplotypes based on Fig 3.**
(TIFF)

**S2 Table. trans*ITS* genetic profiles. A**. * SNP position reference, *S.haematobium* NCBI (MH014047), J = Kachi Samples, OD = Ile Oluji/Oke Igbo samples, *Sh = S. haematobium*, *Sb = S. bovis*, *Sc = S. curassoni*. Each row represents a phased haplotype that was resolved. **B**. Representative DNA sequence electropherogram profiles at the *transITS* PCR-amplified population for OD samples that either possess alleles consistent with an *Sh* x Sc hybrid genotype (OD9, OD12) or an *Sc* genotype (OD17). Dye peaks denoted by asterisks at positions 671, 726, 776, 846 based on the published sequence (GenBank accession no. MH014047) clearly depict the presence of two nucleotides with similar peak heights, whereas only a single, homozygous "A" nucleotide was resolved at position 19.
(TIFF)

## Acknowledgments

We would like the thank the Natural History Museum, London for the genetic material provided via the Schistosomiasis Collection at the Natural History Museum (SCAN) archive, it was curated and collected in a collaborative effort, and we acknowledge the support and generosity of the partnerships involved in the collections, particularly in relation to the endemic countries involved, and past colleagues that maintained the live material. The Schistosomiasis Collection at the Natural History Museum (SCAN) was funded by the Wellcome Trust (grant 104958/Z/14/Z). We would also like to thank the Schistosome Resource Centre of the Biomedical Research Institute (sponsored by NIAID, NIH), and in particular, Dr. Margaret Mentink-Kane.

## Author Contributions

**Conceptualization:** Oluwaremilekun G. Ajakaye.

**Data curation:** Oluwaremilekun G. Ajakaye, Michael E. Grigg.

**Formal analysis:** Oluwaremilekun G. Ajakaye.

**Investigation:** Joshua B. Balogun, Oyetunde T. Oyeyemi.

**Methodology:** Michael E. Grigg.

**Resources:** Michael E. Grigg.

**Supervision:** Michael E. Grigg.

**Validation:** Michael E. Grigg.

**Visualization:** Oluwaremilekun G. Ajakaye.

**Writing – original draft:** Oluwaremilekun G. Ajakaye, Elisha E. Enabulele.

**Writing – review & editing:** Michael E. Grigg.

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
