## [Decision Letter · Decision Letter 0]

25 Aug 2023

Dear Dr. Grigg,

Thank you very much for submitting your manuscript "Extant interspecific hybridization among trematodes within the Schistosoma haematobium species complex in Nigeria" for consideration at PLOS Neglected Tropical Diseases. As with all papers reviewed by the journal, your manuscript was reviewed by members of the editorial board and by several independent reviewers. The reviewers appreciated the attention to an important topic. Based on the reviews, we are likely to accept this manuscript for publication, providing that you modify the manuscript according to the review recommendations. 

The reviewers have done an excellent job comment on the manuscript. I agree with them that the paper has the potential to represent a really valuable contribution to PLoS NTD but needs some improvement before publication. While I do not insist that you separate results and discussion as reviewer 1 suggests, I agree with that reviewer (and the others) that some additional discussion of the results is needed and that the results could be more clearly and comprehensively presented.

Please take especially seriously the requests for additional clarity about the ethical framework for the study, as this seems to me essential for publication

Sincerely,

James Cotton

Academic Editor

Uriel Koziol

Section Editor

The reviewers have done an excellent job comment on the manuscript. I agree with them that the paper has the potential to represent a really valuable contribution to PLoS NTD but needs some improvement before publication. While I do not insist that you separate results and discussion as reviewer 1 suggests, I agree with that reviewer (and the others) that some additional discussion of the results is needed and that the results could be more clearly and comprehensively presented.

Please take especially seriously the requests for additional clarity about the ethical framework for the study, as this seems to me essential for publication

Reviewer's Responses to Questions

**Key Review Criteria Required for Acceptance?**

**Methods**

-Are the objectives of the study clearly articulated with a clear testable hypothesis stated?

-Is the study design appropriate to address the stated objectives?

-Is the population clearly described and appropriate for the hypothesis being tested?

-Is the sample size sufficient to ensure adequate power to address the hypothesis being tested?

-Were correct statistical analysis used to support conclusions?

-Are there concerns about ethical or regulatory requirements being met?

Reviewer #1: Generally the methods need more detail added. 

Line 224 – Ethics statement. There is no detail regarding what happened to the participants if they were found to be schistosomiasis positive. 

Line 246-252 – it is not clear how the urines were identified as positive. You say positive urines were filtered for egg collection. From reading the text it is not clear if this was used to identify positive samples. Also, say how much urine was filtered. 

Line 254-256 – this information is not clear. Firstly, how was the filter/eggs treated before extraction as they were preserved in Ethanol. You say five Schistosoma species were used as positive controls – do you mean DNA from these species? Also, clearly state where the controls came from (source and origin) and use references. 

Line 256-257 – was the filter removed from the lysate before using the Qiagen columns? Also, how much water was used for the elutions? 

Line 259 – is transITS used in other papers? – it is typically referred to as ITS1+2 – changing the name may cause confusion. Also, when you say “isolates” do you mean samples? 

Line 261-268 – as you are using primers that you have newly developed you should explain how these have been developed and tested. 

Line 263 – it is not clear why you are designed a S. bovis specific PCR as you are also using cox1 and ND5 data (pan genus) that is sequenced. You also do not say which molecular target your S. bovis specific PCR is for. 

Line 265 – the new target “Bravo-Figey domain needs some information. How was this identified and how were the primers designed? 

Line 268-269 – you say you sequenced DNA from the 5 controls – which targets were sequenced? 

Line 270 – you do not described the PCR’s for the cox1 and ITS. 

Line 276-284 – you need to add more information to your analyses. E.g. what type of phylogenetic analysis was done? Which molecular targets were analysed by which method. What are your fragment lengths that were analysed. Was the ITS data analysed looking at the species specific SNP regions?

Reviewer #2: Minor and some more major revisions have been detailed below. The study artciulates testable hp and objectives, but details about the study design are missing (and discrepancy between the age of the population surveyed is present throughout the text). Concerns about statistical approaches and other clarifications to ensure compliance to ethical requirements have been highlighted in the comments below.

Reviewer #3: Overall, the authors' methodology is sufficient to address the questions posed and consistent with similar studies/publications in the field. There are just a few minor points/suggestions I would like the authors to consider:

1 - The authors describe the agrarian site as being "devoid of livestock", but I think the support of their chosen sites would be improved if this description was made more epidemiologically clear. Not to be glib, but how devoid is "devoid"? At what geographical scale is the study area "devoid" of livestock relative to the average regular movement and water usage patterns of the local human populace? Is there literally no opportunity for shared human-livestock water source usage outside of a rare long-distance trip by a resident? No opportunity for rodent hosts to occupy a geographical hybrid zone between the agrarian area and a neighboring area where livestock ownership is more prevalent?

1b - The way that the authors refer to the study sites would benefit from some more consistent language. At times the sites are referred to by their name and at others by the methodological descriptors (e.g., "pastoral" vs. "agrarian"/"devoid of livestock"). For the sake of clarity and reader efficiency, I would suggest picking a reference and being consistent in its use -- ideally something that relates directly to the study question/methodology.

2 - As currently presented in the methods section, total DNA used for the sequencing was derived from pools of eggs (Line254: "Total DNA was extracted from individual filters..." -- not from individual eggs or miracidia). That is acceptable methodology, but the term "isolate" is consistently used throughout the manuscript to report and discuss results in a manner that could lead a reader to the conclusion that the results deal with individual larval genotypes, rather than profiles of allele presence/absence in a pooled sample (e.g., "isolate J15 was heterozygous" - Line176). Data in Table 2 are presented in such a way that imply a single isolate's genotype or that all individuals in the sampled pool have the same patterns of zygosity. For instance, it is clear that the results for Ondo are consistent with widespread ("universal"?) mitonuclear discordance, but are the authors suggesting that all of the eggs in the pool comprising sample OD1, for example, are heterozygous for S.h. and S.c. alleles at the ITS locus? It may just be a matter of degrees, but that would have a different epidemiological implication than finding that a subset of individual larvae are the product of S.h.xS.c.xS.b. hybridization. If the authors are suggesting that all of the larvae in the OD1 pool are, in fact, S.h.xS.c.xS.b. hybrids, then there should be some inclusion of the assessment of the relative peak height of heterozygous sites in the methodology and/or results that support that conclusion. This is especially true for the results from Kachi, where one could envision the potential for variability of hybrid prevalence. Using relative peak heights to derive an estimate of hybrid prevalence in Kachi could be an interesting addition. But, more importantly, I would urge more clarity/transparency in the presentation of the results. At a minimum, remind the reader early on in the Results section that the data for "isolates" are derived from pools of samples and reference the limitation in the ability to provide an absolute prevalence of the various hybrid combinations.

3 - For several sample pools, analysis of CO1 was reported as "not done" (Table 2). Please clarify whether these were failed PCR/sequencing reactions or whether there was a conscious decision not to collect CO1 data for those specimen and, if so, why not. 

4 - Very minor: in the Materials and Methods please report the source of the positive control materials.

**Results**

-Does the analysis presented match the analysis plan?

-Are the results clearly and completely presented?

-Are the figures (Tables, Images) of sufficient quality for clarity?

Reviewer #1: Results and Discussion

Unfortunately the way the results are written make it very difficult to understand and interpret. More clarity is needed. It would help if the results and discussion were to be separated so the results can clearly be shown without incorporation of the discussion text, which makes it hard to follow. 

Lines 122 and 123 – the use of “aggregate” and “agrarian” is used but this just adds confusion and you do not explain what these terms mean. 

Table 1 – it is not clear what intensity is based on. Was it 10 mls of filtered urine? 

Genetic profiling in general – as your data is based on pooled samples you cannot precisely say what the genetic profiled is for all your samples. You can say you have signal from the different species in the different markers. It is surprising that you did not get cox1 and ND5 sequences from both Sb and Sh in your individual samples? 

Line 137 – this SNP has been identified before – reference Savassi et al., 2020. 

Line 149 – when you say “allele” – do you mean haplotype? 

In the text it would help if you referred to specific samples in the table when you are discussing the paper – a good example of this is in line 154. 

It is surprising when talking about the haplotype diversity for S. haematobium you do not reference Webster et al., 2012. 

Lines 168-174 – much of this is methods and data should be shown for the testing of the markers on the different species. Same for the ND5. 

For the BF markers – you need to explain if this signal is based on SNP’s or a sequence – as it is a protein coding gene it would be expected to be based on a sequence and so it is not clear how you analysed the data when it was heterozygous. Would you not expect natural variation in the BF target? 

Line 195 – the nested configuration was not clear in the methods.

Reviewer #2: The results are clearly presented but modifications and further details are necessary. These have been highlighted in the comments below.

Reviewer #3: Table 1 -- This table is titled as "S. haematobium prevalence..." but prevalence is only 1 of 4 metrics presented and the title could lead to confusion as it stands (or at least require the reader to spend more time deciphering the information). It would be more precise to say that it is "Study population demographics and S. haematobium prevalence", or something comparable. 

Figure 3 -- The color differences should be more apparent and should be tested for color blind friendliness (e.g., via the Coblis [https://www.color-blindness.com/coblis-color-blindness-simulator/] or Color Oracle [https://www.colororacle.org/] tools). The blues and sea green are particularly hard to distinguish. Also, if possible, please work with the angles of the small S. bovis branches to further separate and clarify any forks and assist with resolving the individual circles.

**Conclusions**

-Are the conclusions supported by the data presented?

-Are the limitations of analysis clearly described?

-Do the authors discuss how these data can be helpful to advance our understanding of the topic under study?

-Is public health relevance addressed?

Reviewer #1: The small conclusion is a good overview of the complexed nature of S. haematobium and these hybrid forms and notes the key future steps needed to fully understand the epidemiology of these species complexes. 

I think the paper would benefit from a distict discussion section

Reviewer #2: The public health relevance and the conclusions of the study are effectively addressed.

Reviewer #3: Overall, the conclusions seem to correspond to the results and methodology, though a key alternative explanation to the observed hybrids in the agrarian zone is not presented -- that of ancient introgression instead of active hybridization. The inability to distinguish ancestral introgression from contemporary hybridization with the methodology they are using should be noted. They are assuming that their results indicate contemporary hybridization, while many of the genomic studies they cite only report evidence for ancestral introgression. That limitation should be included as a caveat to the interpretation of the results, and they might also find it helpful to include the recent paper by Berger et al. (Genomic evidence of contemporary hybridization between Schistosoma species, 2022, PLoS Pathog) to lend support to the notion that the results presented here could feasibly be due to contemporary hybridization.

**Editorial and Data Presentation Modifications?**

Reviewer #1: The paper could present important data. However, there are many details missing and the results are not presented clearly. The text should be read carefully to correct mistakes and also to improve clarity.

Reviewer #2: LINES 32-33. The first time a scientific species name is mentioned it should be fully spelled out. The same goes for the genomic markers (and refer to them as ITS and cox1 rather than transITS and CO1). Modify here and throughout the text.

LINE 38. Use Sh x Sb abbreviated since you already specified it above.

LINE 53. May pose rather than poses.

LINES 80-83. This paragraph may be missing a reference, the provided [4] seem to refer to work done in Senegal.

LINE 93. More than failure, those studies highlighted the absence of ongoing hybridization events.

LINES 119-122. Refer

---

## [Editor Report · Decision Letter 1]

4 Jan 2024

Dear Dr. Grigg,

Thank you very much for submitting your manuscript "Extant interspecific hybridization among trematodes within the Schistosoma haematobium species complex in Nigeria" for consideration at PLOS Neglected Tropical Diseases. As with all papers reviewed by the journal, your manuscript was reviewed by members of the editorial board and by several independent reviewers. The reviewers appreciated the attention to an important topic. Based on the reviews, we are likely to accept this manuscript for publication, providing that you modify the manuscript according to the review recommendations. 

The authors have done a good job of addressing most of the review comments, but there are a couple of places where I think more is needed to address the reviewer concerns from the first round of reviews.

The most important thing is to include some caveats around the interpretation of "hybrids" from pools of eggs, and i'm not sure the response to the reviewer #1 and #3 comments is really completely adequate. In particular, finding identical results in two independent PCR reactions may not be so surprising if analysis is from a large pool of individual miracidia. 

I think this also relates to the concerns reviewer #1 has about a lack of discussion section: as at the very least something needs to be said to make sure it is abundantly clear that the restults are (strictly speaking) findings of diversity of markers in pools of miracidia, rather than heterozygosity of individual organisms. I would like to see several sentences explaining this clearly and explaining the haplotype inference so readers who don't delve into the methods are not misled. 

it would also be useful to see some description of what the authors think is going on in these hybrid populations. As schistosomes are obligately sexual in the adult stage, we might expect to see something like Hardy-Weinberg proportions of heterozygous and homozygous markers, but instead these populations appear to be extremely heterozygous. How do they consider this is likely to come about? To my mind, the obvious explanation would be that the F1 hybrids are infertile, but that doesn't tie in with the proposed 3-way hybridisation. I'd be interested to see the author's proposal for the possible biology here, and think that would make some useful discussion.

Some minor comments: 

line 38: could be made clearer that there were only 57 positive samples.. 

line 77: "It is the second most prevalent tropical disease caused by five major species" - this needs re-wording, as it currently implies that a more prevalent disease is caused by the same 5 species. It needs 'and is caused by'.

line 89: There is a formatting error here: [6, 7,] (delete final comma)

line 93: 'resulting in only' would be better as 'at only'

line 121-123. - I tmight be good to say something about how far away the nearest cattle-rearing area is? That might speak to the possibility of transmission mediated through rodents etc. etc.

line 143: Of particular interest, a private SNP at nucleotide 502 (based on GenBank reference sequence MH014047) was identified that was unique to strains circulating in Kachi, this has been previously observed by Savassi et al, (2020) [15] in S. haematobium or S. bovis samples obtained from human infections (Suppl.Table 2).

IT would be useful for readers (and editors) to explicitly point out which SNP in the Savassi et al. paper position 502 here represents, as they seem to be on different co-ordinate systems. I guess this is 556 in the Savassi et al?

line 145: by Savassi et al, (2020) [15] - it would be better 'style' to just say [15]

We thank the reviewer for their perspective, the amplification of both BF and ND5 was robust for all species tested*

line 154: 'that resolved from other human and livestock schistosomes' - doesn't read as quite grammatical to me. I tihnk this needs some extra words? - maybe something like 'resolved as a monophyletic group excluding other human..'

line 217 : hyrid -> hybrid 

line 226: individually genotype miracidia -> I think 'genotype individual miracidia' would be clearer here.

line 264 and 265: S. haematobium should be in italics.

line 268- 269: Statistical analysis was done using the Fisher exact test to compare the prevalence and intensity of infection with demographic variables.” -- Does this make sense? are all such variable categorical?

line 279:

'and an S. bovis worm (extracted from a cow in Nigeria)'

How confident can the authors be that this is a "pure" S. bovis? Does it matter?

in your response to review #2, you refer to the postiive controls as "typed specimens", and while I guess the SCAN specimens were typed by the collectors, you don't say anything about how this was 'typed'.. or is assumed that cattle schistosomes in that area are 'pure' S. bovis.?? Something should be said.

line 280: 'Pure Genomic DNA' - not sure what this means, so would be good to have some explanation?

Lines 316-317 - I think it would be good to at least add a statement about the degree of testing. How many samples were tested of each species? If more than one, was amplification (and sequence data) identical in every case? 

The writing goes a bit wrong at line 337: 

"was implemented using MEGA X based on the best model (lowest Bayesian information criterion score) for each gene marker [29], using Maximum Likelihood method at 1000 bootstrap replicates to ascertain the reliability of the tree branches. Haplotype network was implemented for CO1 sequence data in an R environment using the Pegas PE package [30]."

Sincerely,

James Cotton

Academic Editor

Uriel Koziol

Section Editor

The authors have done a good job of addressing most of the review comments, but there are a couple of places where I think more is needed to address the reviewer concerns from the first round of reviews.

The most important thing is to include some caveats around the interpretation of "hybrids" from pools of eggs, and i'm not sure the response to the reviewer #1 and #3 comments is really completely adequate. In particular, finding identical results in two independent PCR reactions may not be so surprising if analysis is from a large pool of individual miracidia. 

I think this also relates to the concerns reviewer #1 has about a lack of discussion section: as at the very least something needs to be said to make sure it is abundantly clear that the restults are (strictly speaking) findings of diversity of markers in pools of miracidia, rather than heterozygosity of individual organisms. I would like to see several sentences explaining this clearly and explaining the haplotype inference so readers who don't delve into the methods are not misled. 

it would also be useful to see some description of what the authors think is going on in these hybrid populations. As schistosomes are obligately sexual in the adult stage, we might expect to see something like Hardy-Weinberg proportions of heterozygous and homozygous markers, but instead these populations appear to be extremely heterozygous. How do they consider this is likely to come about? To my mind, the obvious explanation would be that the F1 hybrids are infertile, but that doesn't tie in with the proposed 3-way hybridisation. I'd be interested to see the author's proposal for the possible biology here, and think that would make some useful discussion.

Some minor comments: 

line 38: could be made clearer that there were only 57 positive samples.. 

line 77: "It is the second most prevalent tropical disease caused by five major species" - this needs re-wording, as it currently implies that a more prevalent disease is caused by the same 5 species. It needs 'and is caused by'.

line 89: There is a formatting error here: [6, 7,] (delete final comma)

line 93: 'resulting in only' would be better as 'at only'

line 121-123. - I tmight be good to say something about how far away the nearest cattle-rearing area is? That might speak to the possibility of transmission mediated through rodents etc. etc.

line 143: Of particular interest, a private SNP at nucleotide 502 (based on GenBank reference sequence MH014047) was identified that was unique to strains circulating in Kachi, this has been previously observed by Savassi et al, (2020) [15] in S. haematobium or S. bovis samples obtained from human infections (Suppl.Table 2).

IT would be useful for readers (and editors) to explicitly point out which SNP in the Savassi et al. paper position 502 here represents, as they seem to be on different co-ordinate systems. I guess this is 556 in the Savassi et al?

line 145: by Savassi et al, (2020) [15] - it would be better 'style' to just say [15]

We thank the reviewer for their perspective, the amplification of both BF and ND5 was robust for all species tested*

line 154: 'that resolved from other human and livestock schistosomes' - doesn't read as quite grammatical to me. I tihnk this needs some extra words? - maybe something like 'resolved as a monophyletic group excluding other human..'

line 217 : hyrid -> hybrid 

line 226: individually genotype miracidia -> I think 'genotype individual miracidia' would be clearer here.

line 264 and 265: S. haematobium should be in italics.

line 268- 269: Statistical analysis was done using the Fisher exact test to compare the prevalence and intensity of infection with demographic variables.” -- Does this make sense? are all such variable categorical?

line 279:

'and an S. bovis worm (extracted from a cow in Nigeria)'

How confident can the authors be that this is a "pure" S. bovis? Does it matter?

in your response to review #2, you refer to the postiive controls as "typed specimens", and while I guess the SCAN specimens were typed by the collectors, you don't say anything about how this was 'typed'.. or is assumed that cattle schistosomes in that area are 'pure' S. bovis.?? Something should be said.

line 280: 'Pure Genomic DNA' - not sure what this means, so would be good to have some explanation?

Lines 316-317 - I think it would be good to at least add a statement about the degree of testing. How many samples were tested of each species? If more than one, was amplification (and sequence data) identical in every case? 

The writing goes a bit wrong at line 337: 

"was implemented using MEGA X based on the best model (lowest Bayesian information criterion score) for each gene marker [29], using Maximum Likelihood method at 1000 bootstrap replicates to ascertain the reliability of the tree branches. Haplotype network was implemented for CO1 sequence data in an R environment using the Pegas PE package [30]."

Figure Files:

Data Requirements:

Reproducibility:

References

---

## [Editor Report · Decision Letter 2]

11 Mar 2024

Dear Dr. Grigg,

Thank you very much for submitting your manuscript "Extant interspecific hybridization among trematodes within the Schistosoma haematobium species complex in Nigeria" for consideration at PLOS Neglected Tropical Diseases. We are likely to accept this manuscript for publication, providing that you modify the manuscript according to the review recommendations. 

I have taken the role of academic editor for this manuscript, replacing the previous academic editor who is currently unavailable, in order to agilize the peer-review process. After reading the original submission and the revised versions of the manuscript, it is my opinion that you have improved the manuscript significantly, addressing the comments of the reviewers and editor.

I would only like to point out two aspects of the manuscript that may have escaped the previous rounds of review:

- Regarding Table 1: if I understand correctly, the p-values refer to tests that are comparing the two locations (Ile Oluji/Oke Igbo vs Kachi). E.g. both locations are significantly different regarding their total prevalence, but not their infection intensity. In the case of age, it would thus seem that the p-value of < 0.00001 refers to the different age composition of the samples at both locations. However, the main text stages that "Age was significantly associated with infection (p<0.00001)" (l. 132) which does not seem to have been tested. It also says "...as has been reported previously" but does not provide a reference.

The legend to Table 1 says "Prevalence percentages are in parenthesis" - however, this is not the case, the numbers in parentheses are only showing the % for each category of each variable.

- Line 215 "The presence of such hybrid Sb x Sc livestock:human Sh genotypes have been described previously and support a model whereby no real or perceived barrier of genetic hybridization exists between livestock and human species of trematodes within the S. haematobium species complex. " - a reference appears to be missing here.

Please modify the text if required, or reply to these comments if I have not interpreted the table correctly. 

I expect this to be the final round of revision for this manuscript, and it should not require the manuscript to be sent to reviewers.

Sincerely,

Uriel Koziol

Section Editor

I have taken the role of editor for this manuscript, replacing the previous academic editor who is currently unavailable, in order to agilize the peer-review process. After reading the original submission and the subsequent revised versions of the manuscript, it is my opinion that you have improved the manuscript significantly, addressing the comments of the reviewers and editor.

I would only like to point out two aspects of the manuscript that may have escaped the previous rounds of review:

- Regarding Table 1: if I understand correctly, the p-values refer to tests that are comparing the two locations (Ile Oluji/Oke Igbo vs Kachi). E.g. both locations are significantly different regarding their total prevalence, but not their infection intensity. In the case of age, it would thus seem that the p-value of < 0.00001 refers to the different age composition of the samples at both locations. However, the main text stages that "Age was significantly associated with infection (p<0.00001)" (l. 132) which does not seem to have been tested. It also says "...as has been reported previously" but does not provide a reference.

The legend to Table 1 says "Prevalence percentages are in parenthesis" - however, this is not the case, the numbers in parentheses are only showing the % for each category of each variable.

- Line 215 "The presence of such hybrid Sb x Sc livestock:human Sh genotypes have been described previously and support a model whereby no real or perceived barrier of genetic hybridization exists between livestock and human species of trematodes within the S. haematobium species complex. " - a reference appears to be missing here.

Please modify the text if required, or reply to these comments if I have not interpreted the table correctly. 

I expect this to be the final round of revision for this manuscript, and it should not require the manuscript to be sent to reviewers.

Figure Files:

Data Requirements:

Reproducibility:

References

---

## [Editor Report · Decision Letter 3]

22 Mar 2024

Dear Dr. Grigg,

We are pleased to inform you that your manuscript 'Extant interspecific hybridization among trematodes within the Schistosoma haematobium species complex in Nigeria' has been provisionally accepted for publication in PLOS Neglected Tropical Diseases.

Best regards,

Uriel Koziol

Section Editor

Uriel Koziol

Section Editor

---

## [Editor Report · Acceptance letter]

5 Apr 2024

Dear Dr. Grigg,

We are delighted to inform you that your manuscript, "Extant interspecific hybridization among trematodes within the Schistosoma haematobium species complex in Nigeria," has been formally accepted for publication in PLOS Neglected Tropical Diseases.

Best regards,

Shaden Kamhawi

co-Editor-in-Chief

Paul Brindley

co-Editor-in-Chief
